# Self-perception of physical appearance of adolescents and associated factors in Addis Ababa, Ethiopia

**Ziyad Ahmed Abdo**[1]*, **Semira Ahmed Seid**[2], **Aynye Negesse Woldekiros**[2]

1 Ethiopian Ministry of Health, Department of Hygiene and Environmental Health, Addis Ababa, Ethiopia,
2 Ethiopian Ministry of Health, Department of Public Relation and Communication, Addis Ababa, Ethiopia

* ziyadahm1982@gmail.com

## Abstract

### Introduction

Establishing a positive body image is a critical factor for adolescents' physical and mental health, as it determines self-confidence, and sustainable individual growth and development throughout their lives. This reality needs to be supported by evidences generated locally. However, there is a lack of information in this regard in the study setting. Thus, the aim of this study was to assess the self-perception of one's physical appearance and its associated factors among adolescents in Addis Ababa, Ethiopia.

### Methods and materials

A community-based, cross-sectional study design was used to conduct the study. At the end of the multistage sampling procedure, a systematic random sampling technique was employed to select 308 study participants from selected districts. The questionnaire was adapted from previous studies as it was appropriate for local context. The data collectors were trained before the data collection and supervised during the data collection period, and the questionnaire was pretested. Bivariate logistic regression was used to identify candidate variables, and then variables with p<0.2 were taken to multiple logistic regressions to identify independent associated factors. Statistical significance was considered at P <0.05 with adjusted odds ratios calculated at 95% CIs.

### Result

A total of 283 adolescents were participated in the study, with a response rate of 91.9%. The overall good self-perception of one's physical appearance was 48.4% [95% CI = 43.8, 54.1]. Having a BMI between 18.5 kg/m²-24.9 kg/m² (AOR = 2.56; 95% CI: 1.45, 4.54), presence of enough sport fields in the school (AOR = 1.89; 95% CI: 1.09, 3.29), having daily access to internet services (AOR = 1.69; 95% CI: 1.07, 2.94), following Ethiopian movies/cinemas (AOR = 2.46; 95% CI: 1.46, 4.15), and regularly following western movies/cinemas (AOR = 2.0; 95% CI: 1.11, 3.59) were significantly associated with a good self-perception of one's physical appearance.

**Data Availability Statement:** The datasets used and/or analyzed during the current study is uploaded as Supporting Information file.

**Competing interests:** The authors have declared that no competing interests exist.

**Abbreviations:** AOR, Adjusted Odds Ratio; BMI, Body Mass Index; CI, Confidence Interval; COR, Crude Odds Ratio; MUAC, Most Upper Arm Circumference; REC, Research and Ethical Committee; SD, Standard Deviation; SPSS, Statistical Package for Social Science.

## Conclusion

According to this study, only 48.4% of respondents reported a good self-perception of their physical appearance. Adolescents and their family need to work to bring normal body mass index, which in turn will increase good self-perceptions of their physical appearance. Schools and the local administrations need to increase availability and access to enough sport fields for students to strengthen their physical fitness, which increases their good self-perception of their physical appearance.

## Introduction

The problem of body image, self-perception and dissatisfaction with one's appearance is widely discussed in the anthropological and medical literatures [1–4]. Physical care and body image, especially during adolescence, are social issues [5]. Harter (1999) describes self-perception as evaluative judgments of attributes within discrete domains such as physical appearance, cognitive competence, athletic competence, social acceptance and so forth [6, 7]. Thus, self-perception develops through experience in the social and physical environment, interactions with significant others, and individual behavioral traits [8, 9]. As children develop a sense of themselves, interact with the world, and gain experiences, their self-concept is affected [10].

The physical changes that the body undergoes during adolescence have a profound impact on an individual's personal and social identity [11]. Adolescent who experience earlier physical development are more likely to engage in risk-taking behavior than their peers, and those who develop more slowly may be more likely to face bullying [12]. Early-maturing adolescents are at greater risk for delinquency and are more likely to engage in antisocial behaviors, including drug and alcohol use, truancy, and precocious sexual activity compared to their peers [13, 14]. For this reason, the process of forming physical self-perception can be considered inherent to this age and can also influence physical exercise habits [5, 15].

School is an important level of education in which adolescents continue to learn the minimum level of knowledge and skills that every citizen should have [9, 16]. Entering high school is a transition to a new educational level and can be stressful for adolescents since they encounter a new curriculum, possibly new and different friends, new teachers, preparation for an important and life-changing exam, and puberty [16]. Specifically, at the secondary level of education, adolescents acquire problem-solving skills and develop them according to social values and basic skills for functioning in society [3]. In addition, this educational process plays a crucial role in the development of self-perception of their physical appearance in adolescents.

Studies conducted in different part of the world have shown that, the prevalence of body image dissatisfaction in developed countries ranges from 35% to 81% in adolescent females and 16% to 55% in adolescent males [9, 17–19]. Despite data-based evidence from developed countries, little is known about the status of adolescent body image dissatisfaction and related factors in developing countries such as Ethiopia [23].

Several lines of evidence suggest that biological factors such as age, sex, puberty, body composition, and psychological factors such as depression, low self-esteem, and adoption of weight loss strategies are factors for adolescent body image dissatisfaction [17]. Parents also affect adolescent self-perception through emotion, motivation, beliefs, values, expectancies, and behaviors [20]. A study also showed that, self-perception of one's body, weight was the most important characteristics in adolescent [21]. Peers are central to children's psychological and self-image development [20].

Adolescents face many challenges; such as solidifying their personality, accepting physical changes, leaving their families, building norms and moral values, choosing a career, and becoming a contributing member of society [17]. These serious developmental challenges and potential sources of stress can be detrimental to the development of healthy self-perception and positive body image [24]. High self-perception and a positive body image have a positive impact on an individual's quality of life, performance, and relationships. Adolescents who fail to develop a positive body image and self-perception are at risk of depression, other health condition and related personal problems [25].

Although the level of satisfaction with the self-perception of body image among adolescents is a reason for the growing attention of national and international organizations related to education and health promotion, governments, and researchers [19], there are a limited number of studies related to the self-perception of the physical appearance of adolescents, and little is known about the factors related to satisfaction with one's body image. Similarly, to our knowledge, there has been no study done on this issue in Ethiopia, specifically in Addis Ababa. As such, this study intends to provide evidence by assessing the magnitude of self-perception of physical appearance and its associated factors among adolescents in Addis Ababa, Ethiopia.

## Materials and methods

### Study setting and design

A community-based, cross-sectional study design was used to conduct the study. The study was conducted in Addis Ababa, Ethiopia. Addis Ababa is the capital and largest city of Ethiopia, as well as the country's political, economic, cultural and tourism center. It is located in the central part of Ethiopia, close to the equator. Sitting at the foot of mount Entoto at an elevation of 2,355 meters, it covers an area of 527 $km^2$ [22]. The current population of Addis Ababa in 2022 is 5,228,000, with a 4.43% increase from 2021 [23].

### Population and eligibility criteria

The source population of this study was all adolescents in Addis Ababa city who were attending high school. The study population was all adolescents in selected districts and attending high school in selected sub cities. All selected adolescent students in selected districts who were in high school education during the data collection time and in the adolescent age group (10–19 years) were included in the study. Adolescents with chronic health problems, especially those with diabetes, were excluded from the study.

### Sample size determination

The required sample size for this study was calculated using a single population proportion formula with the following assumptions: 95% confidence level, 5% margin of error, and 19.5% proportion of satisfaction with one's physical appearance taken from a previous study performed in Brazil [9].

$$n = \frac{(1.96)^2 \, 0.195(1 - 0.195)}{(0.05)^2} = 280$$

By considering the 10% non-response rate, a total of 308 sample populations were included.

### Sampling techniques

Multistage sampling techniques were used to select study participants. First, Kolfe Keranio and Yeka sub-cities were selected from sub-cities in Addis Ababa using a simple random

sampling system, and then three districts from each of the selected sub cities were nominated using a simple random sampling system. After the number of adolescent students attending high school was identified in each selected district, the study population was assigned proportionally to each district as per the number of adolescent students attending high school in the district. Then, study participants were selected by using a systematic random sampling techniques.

## Study variables

- Dependent Variable: Self-perception of physical appearance. **_Good self-perception_:**—refers to respondents who scored the mean and above the mean value of self-perception of one's physical appearance score. **_Poor self-perception_:**—refers to respondents who scored below the mean value of self-perception of one's physical appearance score [24].

- Independent Variables: Socio-demographic characteristics: Age, sex, class level, etc., Availability and accessibility of physical activities equipment in the school and nearby, Access to media (as source of information), Perception of their own dietary condition. **_Good perception of dietary and related condition_:**—the respondents who scored the mean and above mean value of score of perception of dietary and related condition. **_Low perception of dietary and related condition_:**—the respondents who scored below the mean value of score of perception of dietary and related condition.

## Data collection tools and quality control

Data were collected using a structured questionnaire through an interview-based process. The questionnaire was adapted from previous studies [3, 9, 25], as it was appropriate for local context. The questionnaire was first prepared in English, then translated to Amharic, and then back to English to ensure consistency. The respondents' self-perception of their physical appearance were measured based on 17 items using likert scale measurement. Prior to the actual data collection, the questionnaire was pre-tested, and then adjustments and corrections were made based on the pretest result. To ensure data quality, adequate training and orientation were given to data collectors and supervisors. The completeness and appropriateness of the collected data were checked by supervisors every day and corrected according to the identified problems. In addition, the investigator monitored and evaluated the overall quality of the data collection process. Weight was measured to the nearest 0.1 kg, and height was measured to the nearest 0.1 cm. According to WHO the BMI-for age >2SD: Obese; >1SD: Overweight; 1 - -2SD: Normal; <− 2 SD: Thinness; <− 3 SD: Severe thinness and height-for-age below -2SD is used for stunting [26].

## Data analysis procedure

The data were entered into EPI data version 3.1 and then exported to SPSS version 23 for data management and analysis. Descriptive statistics of frequency, percentage, means, standard deviation, and standard error of mean were calculated. For those likert scale data, the mean value with standard error of mean was calculated, their percentage was calculated based on a mean value explained in operational definition of variables. Logistic regression was employed to assess factors associated with self-perception of one's physical appearance. Accordingly, independent variables with $P<0.2$ during bivariate logistic regression were included in multivariable logistic regression analysis to identify the association between factors and a good self-perception of one's physical appearance. Adjusted odds ratios with 95% confidence intervals

and significance levels of P< 0.05 were used to identify factors associated with a good self-perception of one's physical appearance.

### Ethics approval and consent to participate

Before conducting the study, ethical clearance was obtained from the Addis Ababa health Bureau ethical clearance board and the Addis Ababa educational bureau. Participants' right to self-determination and autonomy was respected, and study participants were given any information they needed verbally and in written prior to self-administration. As such, written consent was obtained from all study participants. Similarly, for those who were under 18 years old, written consent was obtained from their parents. The rights of each respondent who refused to answer a few or all questions were respected. Participation was voluntary, and participants could withdraw from the study at any time without explanation.

## Results

### Socio-demographic characteristics of respondents

A total of 283 adolescents participated in the study, with a response rate of 91.9%. About 144 (50.9%) of respondents were male. The mean age of the respondents was 16.55 ± 1.2 (SD) years. Most of respondents 116(41%), were attending grade 9. More than 92% of respondents had both parents (father and mother) alive. More than 70% monthly income level were less than 18000 ETB (Table 1).

### Anthropometric measurement of respondents

The majority of the respondents, 177(62.5%) were within the normal range (18.5–24.9 kg/m$^2$) of body mass index, while approximately 19 (6.7%) were overweight (Table 2). The mean height of the respondents was 1.65 m with ± 0.095 (SD), while the mean weight of the respondents was 55.8 kg with ± 9.5 (SD).

### Perception of dietary and related condition of the respondents

The overall good perception of dietary and related condition self-assessment of respondents were 55.5% and 44.6% had low perception. Based on the 5 level likert scale measurement of 19 items, the overall mean perception of respondents was 2.9 (±0.02) (Table 3).

### Availability and accessibility of physical activity equipment

Among respondents, approximately 201 (71%) responded that there were regular physical activities in the school they were learning in. The majority, 218 (77%), of respondents responded that there was no gymnasium in the school. A total of 197 (69.6%) respondents had enough football field in the school they were learning in. About 203 (71.7%) respondents said there were sport clubs in the school. Approximately 155 (54.8%) and 145 (51.2%) respondents had regular physical activity in school and outside school, respectively (Table 4).

### Access to media as source of information

About 278 (98.5%) and 201 (71%) respondents had access to television and radio in their homes, respectively. Out of respondents, 241 (85.2%) had access to private mobile phones, while 193 (68.2%) had daily access to internet services. A total of 169 (59.7%) respondents accessed at least two types of social media daily. Approximately 151 (53.6%) and 201 (71%) respondents regularly followed Ethiopian and Western movies, respectively. Similarly, 130

**Table 1. Distribution of socio-demographic characteristics of adolescents in Addis Ababa city, Ethiopia.**

| Characteristics | Category | Frequency | Percent |
|---|---|---|---|
| Type of the school | Governmental | 150 | 53 |
| | Private | 133 | 47 |
| Sex of respondents | Male | 144 | 50.9 |
| | Female | 139 | 49.1 |
| Class level of the respondents | Grade 9 | 116 | 41 |
| | Grade 10 | 82 | 82 |
| | Grade 11 | 50 | 17 |
| | Grade 12 | 35 | 12.4 |
| Age of respondents in years | 14–17 years | 251 | 76 |
| | 18–19 Years | 68 | 24 |
| Is your parent alive? | Both father and mother is alive | 263 | 92.9 |
| | Only father is alive | 6 | 2.1 |
| | Only mother is alive | 12 | 4.2 |
| | Both father and mother is not alive | 2 | 0.7 |
| Do you live with your parent | Living with both parent | 233 | 82.3 |
| | Living with only one parent | 37 | 13.1 |
| | Living with relative | 13 | 4.6 |
| Father educational level | Illiterate | 7 | 2.6 |
| | Read and write | 9 | 3.3 |
| | Elementary | 19 | 7.1 |
| | High school | 84 | 31.2 |
| | Diploma | 40 | 14.9 |
| | > = first degree | 110 | 40.9 |
| Mother educational level | illiterate | 10 | 3.6 |
| | Read and write | 16 | 5.8 |
| | Elementary | 22 | 8.0 |
| | High school | 93 | 33.5 |
| | Diploma | 63 | 22.9 |
| | > = first degree | 72 | 26.2 |
| Mother occupation | Employer of (gov't & private org.) | 74 | 26.9 |
| | Merchant | 65 | 23.6 |
| | Daily laborer | 21 | 7.6 |
| | House wife | 113 | 41.1 |
| | Farmer | 2 | 0.7 |
| Father occupation | Employer of (gov't & private org.) | 136 | 50.6 |
| | Merchant | 121 | 45.0 |
| | Daily laborer | 7 | 2.6 |
| | Farmer | 5 | 1.8 |
| Family monthly income in ETB | < = 18000 | 199 | 70.3 |
| | >18000 | 84 | 29.7 |

**Table 2. Anthropometric measurement of adolescents in Addis Ababa city, Ethiopia.**

| Variables | Category | Frequency | Percent |
|---|---|---|---|
| BMI of respondents | <18.5 kg/m$^2$ | 87 | 30.7 |
| | 18.5–24.9 kg/m$^2$ | 177 | 62.5 |
| | 25–29.9 kg/m$^2$ | 19 | 6.7 |

**Table 3. Mean perception of dietary and related condition of the adolescent in Addis Ababa city, Ethiopia.**

| Characteristics | Mean (SEM) | Percent |
|---|---|---|
| Frightened of being overweight | 4.13(0.076) | 82.6 |
| Aware of the calorie content of foods that I eat | 1.88(0.067) | 37.6 |
| Particularly avoid food with high carbohydrate content | 4.21(0.064) | 0.842 |
| Think burning up calories when I exercise | 2.48(0.088) | 49.6 |
| Other people think that I am too thin | 4.08(0.080) | 81.6 |
| Avoid food with much sugar | 1.67(0.057) | 33.4 |
| Eat diet food | 1.87(0.069) | 37.4 |
| I feel uncomfortable after eating sweets | 4.3(0.068) | 0.86 |
| Engage in dieting behavior | 1.61(0.054) | 32.2 |
| Feel that food control my life | 1.89(0.074) | 37.8 |
| Give too much time and thought to food | 1.89(0.073) | 37.8 |
| Have the urge to vomit after meals | 4.65(0.044) | 93 |
| Avoid eating when I am hungry | 1.49(0.052) | 29.8 |
| Cut my food into pieces | 1.82(0.062) | 36.4 |
| Feel that others would prefer if I ate more | 1.81(0.069) | 36.2 |
| I am preoccupied with the thought with of having fat on my body | 1.72(0.065) | 34.4 |
| Take longer than other to eat my meal | 4.3(0.059) | 86 |
| Display self-control around food | 2.15(0.074) | 43 |
| Feel that others pressure me to eat | 4.24(0.068) | 84.8 |

SEM: Standard error of the mean

(54.1%) and 157 (55.5%) respondents regularly followed Ethiopian and Western music's, respectively (Table 5).

## Level of self-perception of one's physical appearance

The overall good self-perception of one's physical appearance of respondents were 48.4% [95% CI = 43.8, 54.1] (Fig 1).

**Table 4. Availability and accessibility of physical activity equipment for adolescents in, Addis Ababa city, Ethiopia.**

| Characteristics | Possible options | Frequency | % |
|---|---|---|---|
| Availability of regular physical activity in the school they learn in | Yes | 201 | 71 |
| | No | 82 | 29 |
| Gymnasium access in the school | yes | 65 | 23 |
| | No | 218 | 77 |
| Availability of enough sport field in the school they learn in | Yes | 197 | 69.6 |
| | No | 86 | 30.4 |
| Availability enough sport field for sport in your district/near by | Yes | 167 | 59 |
| | No | 116 | 41 |
| Availability of sport club in your school | Yes | 203 | 71.7 |
| | No | 80 | 28.3 |
| Had regular physical activity in the school | Yes | 155 | 54.8 |
| | No | 128 | 45.2 |
| Had regular physical activity outside school | Yes | 145 | 51.2 |
| | No | 138 | 48.8 |

**Table 5. Media access of adolescents in Addis Ababa city, Ethiopia.**

| Characteristics | Possible options | Frequency | Percent |
|---|---|---|---|
| Access to Television in their home | Yes | 278 | 98.2 |
| | No | 5 | 1.8 |
| Access to radio in their home | Yes | 201 | 71 |
| | No | 82 | 29 |
| Access to private mobile phone | Yes | 241 | 85.2 |
| | No | 42 | 14.8 |
| Access to internet daily | Yes | 193 | 68.2 |
| | No | 90 | 31.8 |
| Do use social media (FB, telegram, Instagram, tweeter, you tube) at least two. | Yes | 169 | 59.7 |
| | No | 114 | 40.3 |
| Follow Ethiopian movies/Cinemas regularly | Yes | 151 | 53.6 |
| | No | 132 | 46.6 |
| Follow western movies/Cinemas regularly | Yes | 201 | 71 |
| | No | 82 | 29 |
| Follow Ethiopian music's regularly | Yes | 153 | 54.1 |
| | No | 130 | 45.9 |
| Follow western music's regularly | Yes | 157 | 55.5 |
| | No | 126 | 44.5 |

## Factors associated with self-perception physical appearance

Logistic regression was conducted to see the association of one independent variable with the dependent variables. Variables with P value <0.2 during the bivariate analysis were included in the multivariate logistic regression analysis to see the relative independent effect of each associated variable by controlling confounding variables. Accordingly, after controlling for covariates, BMI of respondents, availability of enough football fields in the school they are learning in, daily access to internet service, following Ethiopian movies regularly, and

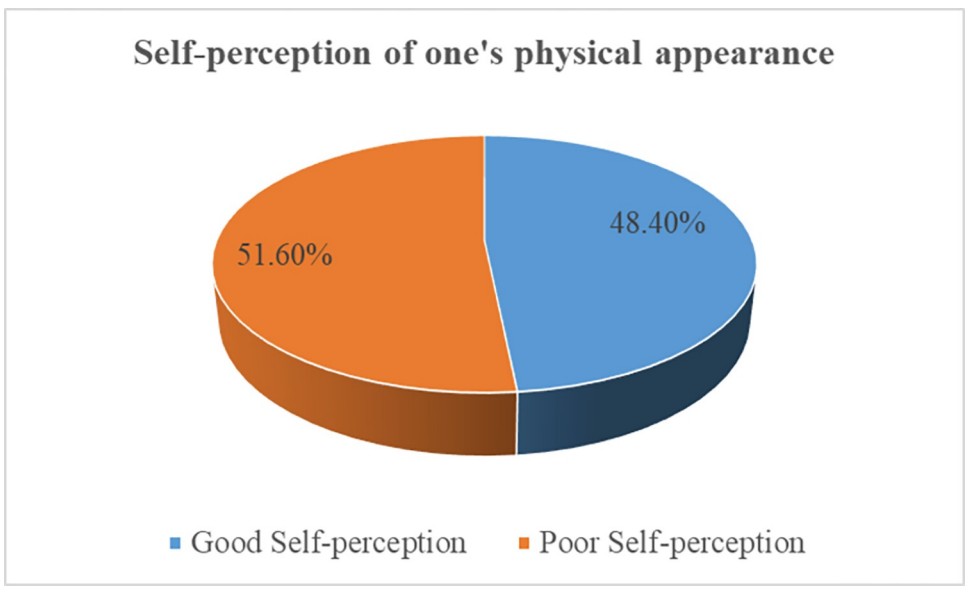

**Fig 1. The level of self-perception of one's physical appearance of adolescents in Addis Ababa city, Ethiopia.**

**Table 6. Factors associated with self-perception of one's physical appearance of adolescents in, Addis Ababa city, Ethiopia.**

| Variables | self-perception of physical appearance | | COR | AOR |
|---|---|---|---|---|
| | **Good** | **Poor** | **[95% of CI]** | **[95% of CI]** |
| | **Freq. (%)** | **Freq. (%)** | | |
| Sex of respondents | | | | |
| Male | 75(52.1) | 69(47.9) | 1.75(1.47, 2.21)* | 1.00(0.59, 1.69) |
| Female | 82(59) | 57(41) | 1 | 1 |
| Age of respondents | | | | |
| 14–17 years | 125(58.1) | 90(41) | 0.64(0.37, 1.11) | 0.67(0.37, 1.22) |
| 18–19 Years | 32(47.1) | 36(52.9) | 1 | 1 |
| BMI of respondents | | | | |
| $<18.5$ kg/m$^2$ | 61(70.1) | 26(29.9) | 1 | 1 |
| 18.5–24.9 kg/m$^2$ | 87(49.2) | 90(50.8) | 2.43(1.4, 4.9)** | 2.56(1.45, 4.54)** |
| 25–29.9 kg/m$^2$ | 9(47.4) | 10(52.6) | 2.61(0.95, 7.16) | 3.58(1.21, 10.56)* |
| Availability of enough sport field in the school you are learning in. | | | | |
| Yes | 117(59.4) | 80(40.6) | 1.68(1.01, 2.80)* | 1.89(1.09, 3.29)* |
| No | 40(46.5) | 46(53.5) | 1 | 1 |
| Having regular physical activities outside their school | | | | |
| Yes | 75(51.7) | 70(48.3) | 1.19(0.46, 1.17) | 1.36(0.80, 2.29) |
| No | 82(59.4) | 56(40.6) | 1 | 1 |
| Having Mobile phone | | | | |
| Yes | 127(52.7) | 114(47.3) | 0.45(0.22, 0.91)* | 1.48(0.62, 3.51) |
| No | 30(71.4) | 12(28.6) | 1 | 1 |
| Daily access to internet service | | | | |
| Yes | 98(50.8) | 95(49.2) | 1.54(1.32, 1.91)* | 1.69(1.07, 2.94)* |
| No | 59(65.6) | 31(34.4) | 1 | 1 |
| Following Ethiopian Movies/Cinemas regularly | | | | |
| Yes | 96(63.6) | 55(36.4) | 2.03(1.26, 3.27)* | 2.46(1.46, 4.15)** |
| No | 61(46.2) | 71(53.8) | 1 | 1 |
| Following Western Movies/Cinemas regularly | | | | |
| Yes | 106(52.7) | 95(74.3) | 1.68(1.2, 2.15)* | 2.00(1.11, 3.59)* |
| No | 51(62.2) | 31(37.8) | 1 | |

*Significance at P<0.05, Significance at P<0.01

following Western music's regularly were significantly associated with adolescent self-perception of one's physical appearance (Table 6).

## Discussion

This study was aimed to assess the level of self-perception of physical appearance and its associated factors among adolescents in the Addis Ababa, Ethiopia. Result of this study show that the level of good self-perception of one's body appearance of adolescents attending high school-level of education in Addis Ababa was 48.4%. The finding of this study is lower than the finding of a similar study conducted in Chile on the self-perception of physical fitness of Chilean adolescent students [3]. Additionally, it is much lower than the finding of a study done in Brazil on body image dissatisfaction among adolescents in public school students, in which only 19.5% of adolescents were dissatisfied with their body image [9]. However, the

finding is higher than that of a study conducted in Romania [27]. The difference might be due to differences in study settings in terms of development, culture, and methodology used to assess self-perception.

Body Mass Index is the most important characteristics in adolescents that determine the self-perception of one's body [4]. The connection between BMI and body image has frequently been discussed in various literary works. Body image relates to people's cognitive and emotional assessments of their body types and proportions, as well as how much value they attach to their outward appearance [28, 29]. As such, the findings of this study also showed that there is a strong association between body mass index and self-perception of one physical appearance in adolescents at P<0.001. Compared to those with lower BMI, those in normal ranges were 2.56 times more likely to have a good perception of their physical appearance. This result is consistent with previous studies [2, 9]. In relation to this, a study conducted among adolescent African American Girls show that, girls with a BMI at or above the 85th percentile were more likely to have greater body image discrepancy and participate in weight control measures than girls with a BMI below the 85th percentile [30].

Having access to a suitable sport field is very important for children's physical development. Physical fitness is related to the ability to perform physical activity and evaluate one's body image [31]. Adolescent's physical self-esteem can be improved by participating in sports. The health benefits of exercise include increased aerobic power, increased muscle strength, and reduced obesity [32, 33]. As such, the results of this study all support this reality. Those who reported that there was enough football field and physical activity in the school had a higher good self-perception of their physical appearance. This result is consistent with a study conducted in Canada [32].

The findings of this study show that there is a positive relationship between using internet services and self-perception of one's physical appearance. This means, compared to adolescents who did not have access to the internet, those who had internet access had a higher good self-perception of their physical appearance. In contrast to this study, many findings showed that there is a negative relationship between access to the internet and self-perception of one's physical appearance [34–36]. Many researchers showed that there is a strong relationship between internet access and self-perception of one's physical [36–39]. Media pressure has been found to be the most powerful factor driving adolescent internalization. Information disseminated through mass media, especially online media, is often unrealistic, and body image research suggests that assimilation leads to discrepancies between actual and ideal body image, leading to lower self-esteem [40].

Movies now make up the majority of media products that people follow [41]. It is the most common way of spending leisure time and has the greatest impact on modern life [42]. Various studies have confirmed many positive effects of movies on children and adolescents. One of them is the self-concept of appearance. This effect depends on the selection of films/movies appropriate for the developmental stages of children and adolescents [43, 44]. As such, the results of this study also showed a strong relationship, P < 0.01, between self-perceptions of one's physical appearance and adherence/following of Ethiopian and Western films. However, some studies show that there is a negative relationship between adherence to cinema and self-perception of one's physical appearance [42, 45].

## Limitation of the study

The major limitation of this study was as nature of cross-sectional study, which may not explain the temporal relationship between the outcome variable and some explanatory variables. As the data type analyzed was quantitative, it may bring less detailed picture about the

problem and its associated factors. In addition, as the study used multistage sampling, there might be some sampling error.

## Conclusion and recommendation

According to this study, only 48.4% of respondents reported a good self-perception of their physical appearance. The results also show that; respondents' BMI, availability of enough sport fields in the school they are learning in, daily access to internet service, following Ethiopian movies/cinemas regularly, following western music regularly are factors that significantly associated with good self-perception of one's physical appearance. Adolescents and their families need to work to achieve normal weight and height to develop a normal body mass index, which strengthens self-perceptions of their physical appearance. Schools and youth centers need to increase access to enough sport fields to strengthen their physical fitness, which increases their self-perception of their physical appearance. Future research should consider well-organized studies that include many variables that enable the determination of underlying causes of poor self-perception of one's physical appearance.

## Supporting information

**S1 Data.**
(SAV)

## Acknowledgments

Our special thanks go to the Addis Ababa city health bureau, all schools included in the study for their willingness to give me supportive letters and important information for my work. Lastly, we would like to thank all participants included in the study for their willingness to participate.

## Author Contributions

**Conceptualization:** Ziyad Ahmed Abdo, Semira Ahmed Seid.

**Data curation:** Ziyad Ahmed Abdo, Aynye Negesse Woldekiros.

**Formal analysis:** Ziyad Ahmed Abdo, Aynye Negesse Woldekiros.

**Funding acquisition:** Ziyad Ahmed Abdo.

**Investigation:** Ziyad Ahmed Abdo, Aynye Negesse Woldekiros.

**Methodology:** Ziyad Ahmed Abdo.

**Project administration:** Ziyad Ahmed Abdo, Semira Ahmed Seid.

**Resources:** Ziyad Ahmed Abdo.

**Software:** Ziyad Ahmed Abdo, Semira Ahmed Seid, Aynye Negesse Woldekiros.

**Supervision:** Ziyad Ahmed Abdo, Semira Ahmed Seid.

**Validation:** Ziyad Ahmed Abdo.

**Visualization:** Ziyad Ahmed Abdo.

**Writing – original draft:** Ziyad Ahmed Abdo.

**Writing – review & editing:** Ziyad Ahmed Abdo, Semira Ahmed Seid, Aynye Negesse Woldekiros.

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
