## [Decision Letter · Decision Letter 0]

26 Jul 2022

PONE-D-22-00860Self-perception of physical appearance of adolescents and associated factors in Addis Ababa, EthiopiaPLOS ONE

Dear Dr. ABDO,

Thank you for submitting your manuscript to PLOS ONE. After careful consideration, we feel that it has merit but does not fully meet PLOS ONE’s publication criteria as it currently stands. Therefore, we invite you to submit a revised version of the manuscript that addresses the points raised during the review process.

Please note that we have only been able to secure a single reviewer to assess your manuscript. We are issuing a decision on your manuscript at this point to prevent further delays in the evaluation of your manuscript. Please be aware that the editor who handles your revised manuscript might find it necessary to invite additional reviewers to assess this work once the revised manuscript is submitted. However, we will aim to proceed on the basis of this single review if possible. 

Your manuscript has been assessed by an expert reviewer, whose comments are appended below (please consult the attached document to view the highlighted sections mentioned by the reviewer). The reviewer has highlighted concerns about several aspects of the methodology and data analysis. Please ensure you respond to each point carefully in your response to reviewers document, and modify your manuscript accordingly.

We look forward to receiving your revised manuscript.

Kind regards,

Joseph Donlan

Editorial Office

PLOS ONE

Journal Requirements:

2 Thank you for stating the following financial disclosure:

 “NO”

a)        Please clarify the soures of funding (financial or material support) for your study. List the grants or organizations that supported your study, including funding received from your institution.

5. Thank you for submitting the above manuscript to PLOS ONE. During our internal evaluation of the manuscript, we found significant text overlap between your submission and the following previously published works, some of which you are an author.

- https://files.eric.ed.gov/fulltext/EJ814480.pdf

- https://cyberleninka.org/article/n/1121227

- https://worldwidescience.org/topicpages/r/reader+self-perception+scale.html

- https://archive.org/stream/ERIC_ED559714/ERIC_ED559714_djvu.txt

- https://moam.info/descarga-del-numero-completo-en-pdf-nutricion-hospitalaria_5a2b4a881723dd76520e4888.html

- https://hrcak.srce.hr/file/245839

- https://onlinelibrary.wiley.com/doi/abs/10.1111/jora.12451

Please revise the manuscript to rephrase the duplicated text, cite your sources, and provide details as to how the current manuscript advances on previous work. Please note that further consideration is dependent on the submission of a manuscript that addresses these concerns about the overlap in text with published work.

Reviewers' comments:

Reviewer's Responses to Questions

**Comments to the Author**

1. Is the manuscript technically sound, and do the data support the conclusions?

Reviewer #1: Partly

2. Has the statistical analysis been performed appropriately and rigorously? 

Reviewer #1: No

3. Have the authors made all data underlying the findings in their manuscript fully available?

Reviewer #1: Yes

4. Is the manuscript presented in an intelligible fashion and written in standard English?

Reviewer #1: No

5. Review Comments to the Author

Reviewer #1: Review manuscript

Research title: Self-perception of physical appearance of adolescents and associated factors in Addis Ababa, Ethiopia

General comment

The researcher was selected a very interesting title that not well studied in developing setting. However, the study was not good methodologically and may not be replicable. Through the document the way of scientific writing needs improvement. The research manuscript need a major revision

Abstract

Introduction: The clear research gap is not clearly stated. The researcher describes the importance of positive body image for physical and mental health. However, the research what is unknown and what the current research adds a new knowledge are not well described.

I would suggest remove the highlighted words from the aim of the study “Therefore, the aim of this study was to assess the level of self-perception of physical appearance and its associated factors among adolescents in Addis Ababa, Ethiopia.”

Method and material

The researcher did not describe how the multistage sampling was done. Its needs a summary of report for the multistage sampling in abstract section because the reader does not understand the sampling units were schools, or districts, or others. It is also better to describe what variables are measured and how before the tell us the analysis. For example, self-perception is measured using structured questionnaire (it is good to mention whether it is validated tool or tools were developed for this study) and describe how the data was collected such as using ‘self-administered’ or ‘interview’. Some other detail information can be presented in the methods section of the manuscript. I also suggest to define the self-perception classification if it has. For example, the researcher said the “over all good self-perception”, what does it mean? The researcher also reported there was also said the good self-perception was low in the conclusion section which need definition in the method section. Indeed, summary of the important methods section needs further improvement.

Results:

Having BMI between 18.5 kg/m2-24.9 kg/m2 (AOR=2.56; 95% CI: 1.45, 4.54), presence of enough sport fields in the school (AOR=1.89; 95% CI: 1.09, 3.29), having daily access to internet services (AOR=1.69; 95% CI: 1.07, 2.94), following Ethiopian movies/cinemas (AOR=2.46; 95% CI: 1.46, 4.15), and regularly following Western movies/cinemas (AOR=2.0; 95% CI: 1.11, 3.59) were significantly associated with self-perception of one’s physical appearance

Does it associated with …good self-perception low self-perception? make it clear or you may say “had statistically significant positive association with self-perception of one’s appearance’

Conclusions: the researcher need to know the nature of a cross-sectional study to determine the associated factors. Thus, researcher must be curious on the factors associated with self-perception which does not mean a causes. There might be various factors that may also associated with self-perception, for example the economy, the social factors, psychological factors, and other factors. The researcher made a shallow recommendation on how the current knowledge is used.

The researcher recommends to increase the access to sport fields …not clear. This look a very general recommendation one can do without

Key words ….>keywords

Introduction

Methodology

The study was a community based cross sectional study. Why the study include only adolescent who attend high school education? Would not be easy to make institution (based) cross-sectional study if the participants are only adolescent who attend high school education?

The researcher selects two sub-cities from Addis Ababa, Ethiopia from 10 or now is 11 sub-cities. Do you think that the result of the study generalizable to Addis Ababa?

‘

The researcher did not mention how the two sub-cities selected? was it randomly or purposively?

is the sampling unit was household or school? This is not clear for the reader

how many household or school was included in the study?

It is also not clear how simple random sampling was applied at community based cross-sectional study or was it school based

Sample size

The sample size calculation not clearly described. What was the proportion of 19.5%? was it dissatisfaction on self-perception? Why the researcher uses the proportion 50% which can give the maximum sample size for the cross-sectional study. The researcher also did not describe the adjustment of sample size because of multistage sampling strategy

Variables:

Dependent Variable: Level of self-perception of physical appearance: bring the operational definition here

The researcher is not consistently used terms in relation to self-perception. In the introduction section they used terms ‘dissatisfaction’ and in the method they used ‘good-self-perception’ and ‘low-self-perception’. Why the researcher wants to use the mean score to classify the self-perception as mean can be affected by skewness. Did the researcher checked the normal distribution of the data before they decide to use the mean score classification? The classification term also makes confuse the reader, for example, in the abstract section the researchers said, “good perception of one’s physical appearance was low”. This makes confused with the self-perception classification ‘’Good” and “low”. I would suggest “good” and “poor’ or “high” and “low” or “satisfied” and “dissatisfied”

I would suggest making the operational definition together with the dependent variable.

The researcher did not clearly describe the variables which needs revision

Data collection tools and quality

The source of the tools is not cited. The researchers did not clearly describe the measurements in details. For examples to measure the self-perception of physical how many questions were used and on what type? Was it liker or yes/No answer options

There is a repetition of statement

Prior to the actual data collection, the questionnaire was adjusted and 18 corrected based on the pretest result, and the final questionnaire was translated to Amharic and 19 then back to English to ensure consistency.

“The questionnaire was pretested and modified 25 based on the findings related to clarity, wordings, logical sequence, skip patterns of the questions 26 and resources needed.”

Merge these two statements into one

The researcher did not explained why BMI and MUAC measurement were used together? Study showed BMI and MUAC highly correlated in Adolescent. What was the assumptions behind using this two indices for nutritional assessment

Data analysis

Were the researchers considering any assumptions for the appropriateness of the analysis they used? Why logistics regression analysis? Did the researcher check the assumption for it?

According to WHO BMI-for age (Obese: >2SD; Overweight: >1SD; Normal: 1 - -2SD; Thinness: <− 2 SD; Severe thinness: <− 3 SD) and height-for-age below -2SD is used for stunting. The MUAC for age can also be used classified for adolescent age. The researcher did not clearly describe the BMI and MUAC classification in this regard. This need to be clearly stated with the reference.

The researcher was used the BMI and MUAC classification for adults not for adolescents

Result

Scio-demographic characteristics

What was the base for monthly income classification?

Perception of dietary and related condition of the respondents

According to responses rated based on five Likert scales, namely, 1-strongly disagree, 2- 4 disagree, 3-not sure, 4-agree and 5-strongly agree, the overall mean perception of respondents 5 was 2.9 (±0.02). Based on the definition of good and low perception, approximately 157 (55.5%) 6 respondents had good perception, and 126 (44.6%) had low perception.

The above statement is not clear for reader

6. PLOS authors have the option to publish the peer review history of their article (what does this mean?). If published, this will include your full peer review and any attached files.

Reviewer #1: No

---

## [Author Response · Author response to Decision Letter 0]

24 Oct 2022

Dear Editor, 

Thank you for giving us the opportunity to submit a revised draft of our manuscript titled (Self-perception of physical appearance of adolescents and associated factors in Addis Ababa, Ethiopia) to Plos One journal. We appreciate you and the reviewer for your precious time in reviewing our paper and providing valuable suggestion. It was your valuable and insightful comments that led to possible improvements in the current version. The authors have carefully considered the comments and tried our best to address every one of them. We hope the manuscript after careful revisions meet your high standards. The authors welcome further constructive comments if any. 

Below we have provide the point-by-point responses/clarification to editor and reviewer comments. In addition, all modifications in the manuscript have been highlighted in red in document uploaded named as “Revised Manuscript with Track Changes”. 

1. Comments from Editor: 

Comment 1: Please ensure that your manuscript meets PLOS ONE's style requirements, including those for file naming. 

Response: We had revised the manuscript as per PLOS ONE’s style requirements including the file naming. 

Comment 2: Statement related to financial disclosure. 

Response: The authors received no specific funding for this work. The same statement is included in the manuscript. 

Comment 3: Issue related to data availability.

Response: We provide the baseline data on protocols.io (https://doi.org/10.17504/protocols.io.8epv5j635l1b/v1) 

Comment 4: Your ethics statement should only appear in the Methods section of your manuscript.

Response: The ethical statement is included under method section by removing from declaration part. 

Comment 5: Thank you for submitting the above manuscript to PLOS ONE. During our internal evaluation of the manuscript, we found significant text overlap between your submission and the following previously published works, some of which you are an author.

Response: We have made great effort to replace, Re-write and adjust those overlapping text. 

2. Comments from the reviewer:

Abstract section comments:

Comment 1: Introduction: The clear research gap is not clearly stated. The researcher describes the importance of positive body image for physical and mental health. However, the research what is unknown and what the current research adds a new knowledge are not well described.

I would suggest remove the highlighted words from the aim of the study “Therefore, the aim of this study was to assess the level of self-perception of physical appearance and its associated factors among adolescents in Addis Ababa, Ethiopia.”

Response: Thank you for your valuable comment. We have tried to show the research gap and what the current research add to current knowledge. Moreover, as abstract word count is limited it is impossible to explain more in this part, so that we had tried to explained more in the introduction part. 

Comment 2: The researcher did not describe how the multistage sampling was done. Its needs a summary of report for the multistage sampling in abstract section because the reader does not understand the sampling units were schools, or districts, or others. It is also better to describe what variables are measured and how before they tell us the analysis. For example, self-perception is measured using structured questionnaire (it is good to mention whether it is validated tool or tools were developed for this study) and describe how the data was collected such as using ‘self-administered’ or ‘interview’. Some other detail information can be presented in the methods section of the manuscript. I also suggest to define the self-perception classification if it has. For example, the researcher said the “over all good self-perception”, what does it mean? The researcher also reported there was also said the good self-perception was low in the conclusion section which need definition in the method section. Indeed, summary of the important methods section needs further improvement.

Response: Thank you again for the comment: The summary of report for the multistage sampling is included in the abstract part as per your comment. 

 The variables measured are included in the manuscript under method section subsection study variables.

 The dependent variable are defined under methodology section 

 We had tried to modify the methodology part to make it clearer to the reader by incorporating all your comments. Please see the methodology section of the manuscript. As you know, under abstract section it is difficult to explain every points as this part word count allowed is very limited as per the journal format. 

Comment 3: Having BMI between 18.5 kg/m2-24.9 kg/m2 (AOR=2.56; 95% CI: 1.45, 4.54), presence of enough sport fields in the school (AOR=1.89; 95% CI: 1.09, 3.29), having daily access to internet services (AOR=1.69; 95% CI: 1.07, 2.94), following Ethiopian movies/cinemas (AOR=2.46; 95% CI: 1.46, 4.15), and regularly following Western movies/cinemas (AOR=2.0; 95% CI: 1.11, 3.59) were significantly associated with self-perception of one’s physical appearance

 Does it associated with …good self-perception low self-perception? Make it clear or you may say “had statistically significant positive association with self-perception of one’s appearance

Response: The comment is well received and corrected accordingly in the manuscript. 

Comment 4: Conclusions: the researcher need to know the nature of a cross-sectional study to determine the associated factors. Thus, researcher must be curious on the factors associated with self-perception which does not mean a causes. There might be various factors that may also associated with self-perception, for example the economy, the social factors, psychological factors, and other factors. The researcher made a shallow recommendation on how the current knowledge is used.

 The researcher recommends to increase the access to sport fields …not clear. This look a very general recommendation one can do without 

 Key words ….>keywords

Response: Of course there might be many factors that associated with the self-perception of one’s physical appearance other than that discussed here. However, we explained and discussed the factors depending the result of our analysis. In addition, as you know the recommendation must be depend on the finding of the study. Anyhow we have adjusted the recommendation to some extent. 

 It is not clear about comment of “Key words” We have included the possible keywords in the manuscript. 

Introduction section comments 

Comment 5: It would be good to include a paragraph explaining how self-perception may be influenced by a variety of factors.

 It is inconsistent throughout the document to use terms like 'dissatisfaction' and 'low self-perception'.

Responses: We have included a paragraph explaining how varies factors do affect self-perception of physical appearance.

 In relation to 'dissatisfaction' and 'low self-perception', some literatures use the first, other use the second one. Hence, in the introduction section part of this manuscript, there may be either of them or both because more of the introduction part was extracted from previous studies that might use either of them to explain all most the same issue. However, the finding of this study is depend on the operational definition we had provided in the methodology part which is either Good self-perception or Poor self-perception. 

Comment 6: The study was a community based cross sectional study. Why the study include only adolescent who attend high school education? Would not be easy to make institution (based) cross-sectional study if the participants are only adolescent who attend high school education?

Response: To make this point clear, this study was first planned to be done at institutional (school) level, however, due to covid-19, the schools were closed as part of covid-19 prevention protocol. Then we have changed to community based. So, the sampling unit is household not school. 

Comment 7: The researcher selects two sub-cities from Addis Ababa, Ethiopia from 10 or now is 11 sub-cities. Do you think that the result of the study generalizable to Addis Ababa?

Response: At the time of starting of this work, the sub cities in Addis Ababa were 10. Hence, we have selected only two sub cities due to limitation of available resources. However, as much as possible, we tried to make the sampling representative by using a simple random sampling technique to select the sub cities and districts. 

Comment 8: I also highly recommend a thorough formatting of the paper for publication in the journal. 

Response: We have tried to format the paper according to the journal formatting style. 

Comment 9: The researcher did not mention how the two sub-cities selected? Was it randomly or purposively?

Response: The two sub cities was selected by simple random sampling system. 

Comment 10: Is the sampling unit was household or school? This is not clear for the reader

Response: The sampling unit was household. This study was first planned to be done at institutional (school) level, however, due to covid-19, the schools were closed as part of covid-19 prevention protocol. Then we have changed to community based. So, the sampling unit is household not school. 

Comment 11: How many household or school was included in the study?

Response: 308 households was included.

Comment 12: It is also not clear how simple random sampling was applied at community based cross-sectional study or was it school based 

Response: The comment is well taken. It is typing error. Corrected to Systematic random sampling system. 

Comment 13: The sample size calculation not clearly described. What was the proportion of 19.5%? Was it dissatisfaction on self-perception? Why the researcher uses the proportion 50% which can give the maximum sample size for the cross-sectional study. The researcher also did not describe the adjustment of sample size because of multistage sampling strategy.

Response. Corrected and we had included sample size calculation description. 19.5% is the proportion of satisfaction not dissatisfaction. As you know 50% is recommended if there is no any previous study done on the topic. Thus why we used the available reference. 

Comment 13: Dependent Variable: Level of self-perception of physical appearance: bring the operational definition here.

Response: We had already included operational definition in methodology part. However, as per your comment we have merged with dependent variable. 

Comment 14: The researcher is not consistently used terms in relation to self-perception. In the introduction section they used terms ‘dissatisfaction’ and in the method they used ‘good-self-perception’ and ‘low-self-perception’.

Response: In relation to 'dissatisfaction' and 'low self-perception', some literatures use the first, other use the second one. Hence, in the introduction section part of this manuscript, there may be either of them or both because more of the introduction part was extracted from previous studies that might use either of them to explain all most the same issue. However, the finding of this study is depend on the operational definition we had provided in operational definition in the methodology part which is either Good self-perception or Poor self-perception. 

Comment 15: Why the researcher wants to use the mean score to classify the self-perception as mean can be affected by skewness. Did the researcher checked the normal distribution of the data before they decide to use the mean score classification? 

Responses: We used mean as cut-point, because our data is normally distributed data. 

Comment 16: The classification term also makes confuse the reader, for example, in the abstract section the researchers said, “Good perception of one’s physical appearance was low”. This makes confused with the self-perception classification ‘’Good” and “low”. I would suggest “good” and “poor’ or “high” and “low” or “satisfied” and “dissatisfied” 

Responses: It good and appropriate comment, so, as per your comment we corrected the operational definition as high” and “poor” satisfaction. As such, this operational definition was used to describe the satisfaction level of the finding of the study. 

Comment 17: I would suggest making the operational definition together with the dependent variable.

Response: Comment total accepted, the operational definition merged with dependent variable. 

Comment 18: The researcher did not clearly describe the variables which needs revision 

Response: Comment accepted and revised accordingly. 

Comment 19: The researchers did not clearly describe the measurements in details. For examples to measure the self-perception of physical how many questions were used and on what type? Was it liker or yes/No answer options.

Response: Comment total accepted and corrected accordingly. See Methodology part sub section Data collection tools and quality control which explain the measurments. 

Comment 20: There is a repetition of statement 

 Prior to the actual data collection, the questionnaire was adjusted and 18 corrected based on the pretest result, and the final questionnaire was translated to Amharic and 19 then back to English to ensure consistency.

“The questionnaire was pretested and modified 25 based on the findings related to clarity, wordings, logical sequence, skip patterns of the questions 26 and resources needed.”

Merge these two statements into one

Response: Comment Accepted. Merged accordingly. 

Comment 21: The researcher did not explained why BMI and MUAC measurement were used together? Study showed BMI and MUAC highly correlated in Adolescent. What was the assumptions behind using this two indices for nutritional assessment? 

Responses: The comment is right, different literatures showed that BMI and MUAC measurements are highly correlated. We assumed that the findings of our study may show the research communities another support in addition to available evidences if both measurements are done simultaneously. Besides, the assumption behind using these two indices for nutritional assessment among adolescents proving researchers thoughts such as “The MUAC is useful especially in monitoring severe under nutrition” and “The BMI is a commonly used measurement for screening both underweight and overweight”.

Comment 22: Were the researchers considering any assumptions for the appropriateness of the analysis they used? Why logistics regression analysis? Did the researcher check the assumption for it? 

Response: As our data is categorical data, we used logistics regression analysis which does not require linearity test. We have made same assumptions before starting the analysis. Example we have checked multi-collinearity test and our data is free from it. 

Comment 23: According to WHO BMI-for age (Obese: >2SD; Overweight: >1SD; Normal: 1 - -2SD; Thinness: <− 2 SD; Severe thinness: <− 3 SD) and height-for-age below -2SD is used for stunting. The MUAC for age can also be used classified for adolescent age. The researcher did not clearly describe the BMI and MUAC classification in this regard. This needs to be clearly stated with the reference. This need to be clearly stated with the reference.

Response: We included the BMI classification explanation with reference in the methodology part according to your comment. Please see sub section data collection tools and quality control under methodology part. 

Result section comments 

Comment 24: What was the base for monthly income classification? 

Responses: The classification of income is depend on the mean value of the income level taken as cut-off point. 

Comment 25: According to responses rated based on five Likert scales, namely, 1-strongly disagree, 2- 4 disagree, 3-not sure, 4-agree and 5-strongly agree, the overall mean perception of respondents 5 was 2.9 (±0.02). Based on the definition of good and low perception, approximately 157 (55.5%) 6 respondents had good perception, and 126 (44.6%) had low perception.

 The above statement is not clear for reader

Response: To make it more clear, we re-write it and add a table

---

## [Decision Letter · Decision Letter 1]

12 Dec 2022

PONE-D-22-00860R1Self-perception of physical appearance of adolescents and associated factors in Addis Ababa, EthiopiaPLOS ONE

Dear Dr. ZIYAD AHMED,

Thank you for submitting your manuscript to PLOS ONE. After careful consideration, we feel that it has merit but does not fully meet PLOS ONE’s publication criteria as it currently stands. Therefore, we invite you to submit a revised version of the manuscript that addresses the points raised during the review process.

ACADEMIC EDITOR: Please insert comments here and delete this placeholder text when finished. Be sure to:

 In the abstract section please rephrase this statement "Thus, the aim of study was to generate evidence by assessing self-perception of physical appearance and its associated factors among adolescents in Addis Ababa." as "Thus, the aim of this study was to assess the self-perception of physical appearance and its associated factors among adolescents in Addis Ababa, Ethiopia."

Please check the English language throughout the manuscript document  

We look forward to receiving your revised manuscript.

Kind regards,

Zekariyas Sahile

Guest Editor

PLOS ONE

Journal Requirements:

Reviewers' comments:

Reviewer's Responses to Questions

**Comments to the Author**

1. If the authors have adequately addressed your comments raised in a previous round of review and you feel that this manuscript is now acceptable for publication, you may indicate that here to bypass the “Comments to the Author” section, enter your conflict of interest statement in the “Confidential to Editor” section, and submit your "Accept" recommendation.

Reviewer #2: All comments have been addressed

2. Is the manuscript technically sound, and do the data support the conclusions?

Reviewer #2: Yes

3. Has the statistical analysis been performed appropriately and rigorously? 

Reviewer #2: Yes

4. Have the authors made all data underlying the findings in their manuscript fully available?

Reviewer #2: Yes

5. Is the manuscript presented in an intelligible fashion and written in standard English?

Reviewer #2: Yes

6. Review Comments to the Author

Reviewer #2: The manuscript has improved significantly from the original submission. However, I believe there are some minor comments which needs authors response. So, I kindly provided my comments and sugessions on the attached PDF file. The authors simply click on the comment icon in the original location which the comments are provided.

7. PLOS authors have the option to publish the peer review history of their article (what does this mean?). If published, this will include your full peer review and any attached files.

Reviewer #2: **Yes: **Benyam Seifu

---

## [Author Response · Author response to Decision Letter 1]

23 Dec 2022

Dear Editor, 

Thank you for giving us the opportunity to submit a revised draft of our manuscript titled (Self-perception of physical appearance of adolescents and associated factors in Addis Ababa, Ethiopia) to Plos One journal. We appreciate you and the reviewer for your precious time in reviewing our paper and providing valuable suggestion. It was your valuable and insightful comments that led to possible improvements in the current version. The authors have carefully considered the comments and tried our best to address every one of them. We hope the manuscript after careful revisions meet your high standards. The authors welcome further constructive comments if any. 

Below we have provide the point-by-point responses/clarification to editor and reviewer comments. In addition, all modifications in the manuscript have been highlighted in red in document uploaded named as “Revised Manuscript with Track Changes”. 

1. Comments from Academic Editor: 

Comment 1: In the abstract section please rephrase this statement "Thus, the aim of study was to generate evidence by assessing self-perception of physical appearance and its associated factors among adolescents in Addis Ababa." as "Thus, the aim of this study was to assess the self-perception of physical appearance and its associated factors among adolescents in Addis Ababa, Ethiopia." 

Response: The comment is well accepted and corrected accordingly. 

Comment 2: Please check the English language throughout the manuscript document 

Response: The authors received no specific funding for this work. The same statement is included in the manuscript. 

2. Comments from the reviewer:

Abstract section comments:

Comment 1: Methods and Materials: (Logistic regression) need paraphrasing to make it more clearly for the readers. Write it in detail because some readers only read the abstract. 

Response: Thank you for your valuable comment. We have tried to make clearer by adding some explanation. 

Comment 2: Result: Add response rate 

Response: The comment is well accepted and added accordingly. 

Comment 3: Conclusion: Compared to Local findings or from global studies. Such conclusion need careful overview. 

Response: We have tried to conclude according to your suggestion. 

Comment 4: Conclusion: Schools and the local administrations need to increase access to enough sport fields for students to strengthen their physical fitness, which increases their good self-perception of their physical appearance. (How this recommendation is drawn from the result is not clear). 

Response: The recommendation is drown because the availability and accessibility of enough sport fields in the school was one of the determinant factor for good self-perception of one’s physical appearance according to the finding of this study. 

Introduction section comments: 

• All specific comments under introduction parts were well addressed and the changes done is highlighted in the document with track change.

Method and Material section Comments: 

Comment 1: Study setting and design: In what year does the specified number of Addis Ababa population.

Response: The comment is corrected by including the current (2022) population size of Addis Ababa. 

Comment 2: Since you have used a multi-stage sampling it is expected to consider design effect. If it is not possible to a correct this at this stage, I suggest you can modify the study population, otherwise with sampling it is difficult to generalize. 

Response: We consider the population as homogeneous. So that, using the calculated sample size without adding design effect may not significantly affect the generalizability of the result. Also due to resource constraint we decided to conduct with this sample size. 

Comment 3: Do you have any reference to use mean score as a cut of point? If so please indicate here 

Response: We have provided a reference whether it is possible to use mean as cutoff point or not. 

 When the data analyzed is normally distributed it is possible to use mean as cut off point. Since our data is likert scale data and in addition it is normally distributed, we had used mean as cut off point. 

Comment 4: There are different logistic regressions please clearly indicate which one is it, am guessing it binary logistic regression so explain clearly.

Response: We have added explanation to make it clearer. 

Comment 5: Determinant and associated factors are different things, please be consistent with associated factors since your design didn’t allow to show cause and effect relationship.

Response: The comment is well accepted and corrected accordingly. 

Result section Comments: 

Comment 1: How 18,000 is used as cut off point

Response: We have used a mean value (18,000) as cut off point. Many published paper were used this cut off points. 

Comment 2: Is there any observation done by the researchers to confirm the Availability and accessibility of physical activity equipment?

Response: The finding didn’t supported by observation. Which is limitation of our study. 

Comment 3: Can you explain the difference of the effects between western and local movies? You can consider explain why this two variables asked differently on the discussion section if not already stated

Response: There is great difference between the two, which follow different dressing, acting etc. styles that motivate, attract the adolescent to follow the same style. So that the adolescent compare their style with them. 

In addition to specific comments, generally we have made to improve the paper to meet high standard of the journal.

---

## [Editor Report · Decision Letter 2]

3 Jan 2023

PONE-D-22-00860R2Self-perception of physical appearance of adolescents and associated factors in Addis Ababa, EthiopiaPLOS ONE

Dear Dr. Ziyad Ahmed

Thank you for submitting your manuscript to PLOS ONE. After careful consideration, we feel that it has merit but does not fully meet PLOS ONE’s publication criteria as it currently stands. Therefore, we invite you to submit a revised version of the manuscript that addresses the points raised during the review process.

In abstract and result sections

Are the confidence intervals "for overall good self-perception of one’s physical appearance was 48.4% [95% CI=48.8, 541]" correct? Please check this.

Following the discussion, the author needs to mention the limitations of the study. Are there any limitations to this study? The reviewers' comments may assist you in writing these limitations.

Please check the consistency of the font size, for example, the confidence interval in the abstract section

Please also check the manuscript is prepared according to the requirement of PLOS one guideline. Manuscript that is not prepared according to the PLOS ONE guideline may delay for publication 

We look forward to receiving your revised manuscript.

Kind regards,

Zekariyas Sahile

Guest Editor

PLOS ONE
---

## [Author Response · Author response to Decision Letter 2]

5 Jan 2023

Dear Editor, 

Thank you for giving us the opportunity to submit a revised draft of our manuscript titled (Self-perception of physical appearance of adolescents and associated factors in Addis Ababa, Ethiopia) to Plos One journal. We appreciate you and the reviewer for your precious time in reviewing our paper and providing valuable suggestion. It was your valuable and insightful comments that led to possible improvements in the current version. The authors have carefully considered the comments and tried our best to address every one of them. We hope the manuscript after careful revisions meet your high standards. The authors welcome further constructive comments if any. 

Below we have provide the point-by-point responses/clarification to editor comments. In addition, all modifications in the manuscript have been highlighted in red in document uploaded named as “Revised Manuscript with Track Changes”. 

Comments from Editor: 

Comment 1: Are the confidence intervals "for overall good self-perception of one’s physical appearance was 48.4% [95% CI=48.8, 541]" correct? Please check this

Response: It is typing error, and the comment is well accepted and corrected accordingly. 

Comment 2: Following the discussion, the author needs to mention the limitations of the study. Are there any limitations to this study? The reviewers' comments may assist you in writing these limitations. 

Response: The comment is well accepted. We included the limitation of the study in the manuscript. 

Comment 3: Please check the consistency of the font size, for example, the confidence interval in the abstract section

Response: Thank you for your valuable comment. The font size was checked throughout the document and corrected as per the journal guideline. 

Comment 4: Please review your reference list to ensure that it is complete and correct.

Response: Checked thoroughly. 

Comment 5: While revising your submission, please upload your figure files to the Preflight Analysis and Conversion Engine (PACE) digital diagnostic tool

Response: Suggestion accepted and uploaded accordingly.

---

## [Editor Report · Decision Letter 3]

6 Jan 2023

PONE-D-22-00860R3Self-perception of physical appearance of adolescents and associated factors in Addis Ababa, EthiopiaPLOS ONE

Dear Dr. Ziyad Ahmed 

Thank you for submitting your manuscript to PLOS ONE. After careful consideration, we feel that it has merit but does not fully meet PLOS ONE’s publication criteria as it currently stands. Therefore, we invite you to submit a revised version of the manuscript that addresses the points raised during the review process.

In the discussion section line 234, the percentage (48.6%) for good self-perception of one's body appearance is not consistent with the abstract and results. Please also check other inconsistency through results which may delay/affect the acceptance of the manuscript. 

The results of Table 3 are unclear when it comes to perceptions of dietary and related conditions. How was the percentage calculated for the Likert scale? The analysis section should also describe this. Is column 2 indicating mean (SD)? which is not clear 

Researchers used a multistage sampling, which could have higher sampling error than simple random sampling. The study should reported in the  limitation 

We look forward to receiving your revised manuscript.

Kind regards,

Zekariyas Sahile

Guest Editor

PLOS ONE
---

## [Author Response · Author response to Decision Letter 3]

15 Jan 2023

Dear Editor, 

Thank you for giving us the opportunity to submit a revised draft of our manuscript titled (Self-perception of physical appearance of adolescents and associated factors in Addis Ababa, Ethiopia) to Plos One journal. We appreciate you and the reviewer for your precious time in reviewing our paper and providing valuable suggestion. It was your valuable and insightful comments that led to possible improvements in the current version. The authors have carefully considered the comments and tried our best to address every one of them. We hope the manuscript after careful revisions meet your high standards. The authors welcome further constructive comments if any. 

Below we have provide the point-by-point responses/clarification to editor comments. In addition, all modifications in the manuscript have been highlighted in red in document uploaded named as “Revised Manuscript with Track Changes”. 

Comments from Editor: 

Comment 1: In the discussion section line 234, the percentage (48.6%) for good self-perception of one's body appearance is not consistent with the abstract and results. Please also check other inconsistency through results which may delay/affect the acceptance of the manuscript. 

Response: The correct percentage is 48.4%, it is typing error, and the comment is well accepted and corrected accordingly. Other inconsistence also checked through the document. 

Comment 2: The results of Table 3 are unclear when it comes to perceptions of dietary and related conditions. How was the percentage calculated for the Likert scale? The analysis section should also describe this. Is column 2 indicating mean (SD)? Which is not clear. 

Response: The comment well accepted. We included operational definition in the variable definition in the methods and materials section, under study variable sub section. Similarly, we added some explanation under data analysis procedure to make clearer. 

Comment 3: Researchers used a multistage sampling, which could have higher sampling error than simple random sampling. The study should reported in the limitation. 

Response: The comment is well accepted, and included in the limitation section.

---

## [Editor Report · Decision Letter 4]

18 Jan 2023

Self-perception of physical appearance of adolescents and associated factors in Addis Ababa, Ethiopia

PONE-D-22-00860R4

Dear Dr. Abdo

We’re pleased to inform you that your manuscript has been judged scientifically suitable for publication and will be formally accepted for publication once it meets all outstanding technical requirements.

Kind regards,

Zekariyas Sahile

Academic Editor

PLOS ONE
---

## [Editor Report · Acceptance letter]

19 Jan 2023

PONE-D-22-00860R4 

Self-perception of Physical Appearance of Adolescents and Associated Factors in Addis Ababa, Ethiopia 

Dear Dr. Abdo:

I'm pleased to inform you that your manuscript has been deemed suitable for publication in PLOS ONE. Congratulations! Your manuscript is now with our production department. 

Kind regards, 

on behalf of

Dr. Zekariyas Sahile 

Academic Editor

PLOS ONE